# Using social capital to address youth sexual and reproductive health and rights in disaster preparedness and response: A qualitative study highlighting the strengths of Pacific community organisations and networks

**Nabreesa Murphy**[1,2,3]*, **Peter Azzopardi**[4], **Kathryn Bowen**[1,2,3], **Phoebe Quinn**[2,3,5], **Tamani Rarama**[6], **Akanisi Dawainavesi**[7], **Meghan A. Bohren**[1]

1 Gender and Women's Health Unit, Centre for Health Equity, School of Population and Global Health, University of Melbourne, Carlton, VIC, Australia, 2 Melbourne Climate Futures, Centre for Health Equity, School of Population and Global Health, University of Melbourne, Carlton, VIC, Australia, 3 Institute for Advanced Sustainability Studies, Potsdam, Germany, 4 Department of Paediatrics, Burnet Institute, South Australian Health and Medical Research Institute, Global Adolescent Health Group, Maternal Child and Adolescent Health Program, AND Adolescent Health and Wellbeing Program, Aboriginal Health Equity Theme, University of Melbourne, Melbourne, Australia, 5 Child & Community Wellbeing Unit, Centre for Health Equity, School of Population and Global Health, The University of Melbourne, Melbourne, Australia, 6 Fiji Youth Sexual and Reproductive Health and Rights Alliance (FYSA), Suva, Fiji, 7 Pacific Sub-Regional Office for International Planned Parenthood Federation (IPPF), Suva, Fiji

* nabreesa@student.unimelb.edu.au

## Abstract

In the Pacific region, youth sexual and reproductive health and rights (SRHR) are strongly influenced by sociocultural and structural factors, which limit access to SRHR information and services for youth. As climate-related disasters intensify in the Pacific, existing challenges to youth SRHR may increase the risk of worse SRHR experiences and outcomes for youth before, during and after disasters. Community-based models of SRHR service provision models increase accessibility for youth in non-disaster times, but there is limited evidence of how community organisations address youth SRHR in disaster contexts. We conducted qualitative interviews with 16 participants from community organisations and networks in Fiji, Vanuatu, and Tonga following the 2020 Tropical Cyclone (TC) Harold. Guided by the Recovery Capitals Framework (natural, built, political, cultural, human, social, and financial capitals), we explored how community organisations addressed challenges to facilitate access to youth SRHR information and services. Social capital in the form of peer networks and virtual safe spaces was used to navigate challenges in political, financial, and natural capitals. Existing relationships and trusted collaborations were crucial to address cultural taboos related to youth SRHR. Previous experiences of disasters and knowledge of contexts enabled participants to provide sustainable solutions to identified SRHR needs. The work conducted by community organisations and networks pre-disaster made it easier to identify and address youth SRHR risks following disasters. Our research offers a unique perspective into how social capitals were used to mitigate challenges to youth SRHR across natural, human, financial, cultural, built, and political capitals. Findings offer important

**Data Availability Statement:** Data collection was through individual semi-structured qualitative interviews. The interview guide is attached as a supplementary document (Appendix A). The data/interview transcripts are not publicly available due to the need to protect participant confidentiality. De-identified excerpts from transcripts may be made available on reasonable request. Requests can be directed to the University of Melbourne Centre for Health Equity (che-enquiry@unimelb.edu.au).

**Funding:** This work was supported by the University of Melbourne (Human Rights Scholarship to NM, Population Health Investing in Research Student's Training [PHIRST] to NM, Dame Kate Campbell Fellowship to MAB), and the Australian Research Council (Discovery Early Career Researcher Award [DE200100264] to MAB). The funders had no role in study design, data collection and analysis, decision to publish, or preparation of the manuscript.

**Competing interests:** I have read the journal's policy and the authors of this manuscript have the following competing interests: PA is an editor for PLOS Global Public Health, but had no editorial role on this paper. AD is a humanitarian practitioner with the Pacific Regional Office for International Planned Parenthood Federation (IPPF) based in Fiji. The authors declare no additional competing interests.

opportunities to invest in existing community strengths, for transformative action to advance the SRHR of Pacific youth.

# 1 Introduction

Climate change is a threat multiplier, increasing risks to health and contributing to worsening humanitarian crises [1,2]. Climate impacts are inequitable, with countries and communities who contribute the least to the climate crisis disproportionately facing increasing disaster risks that threaten their health and human rights [1]. Disaster risk is defined as the consequence of interactions between a hazard (e.g., a cyclone) and the characteristics that make people and places vulnerable and exposed (eg health status, access to physical or financial resources) [3]. For communities, vulnerability and exposure to disasters are often created or magnified through institutional failures, due to human decisions and actions that exacerbate underlying social and political inequities [4,5]. This means that a hazard triggers a disaster because it disproportionately impacts the lives and livelihoods of people who face increased risks due to marginalisation and inequitable access to resources, information and services [6]. Consequently, disasters disproportionately affect individuals and communities who encounter structural inequities and injustices.

## 1.1 Sexual and reproductive health and rights in disasters

Sexual and reproductive health and rights (SRHR) are essential to sustainable development, as they are inextricably linked with gender equity and influence health and well-being throughout the life course. The inequitable burden of climate change and disasters have important implications for those who face barriers to realising their SRHR. For instance, in humanitarian settings, adolescents, male survivors of sexual violence, people with disabilities, and people with sexual and gender diversity continue to face significant obstacles in accessing SRHR information and services [7]. Existing norms, social attitudes and traditions intersect with other forms of inequity and result in increased risks to SRHR for women and sexual and gender minorities in particular [8].

Pacific Island Countries are facing increasing risks of climate-related disasters related to sea level rise, rapidly changing weather patterns, and extreme weather events [9,10]. As a region with a large youth population, the increasing frequency of disasters in the Pacific has significant implications for the SRHR of Pacific youth [10–13]. Pacific youth SRHR are strongly influenced by sociocultural and structural factors, including persistent gender inequity, expectations about appropriate sexual behaviour, and stigma and misconceptions around gender, sexuality, and reproduction [14–19]. Menstrual health taboos, and violence and discrimination against people with diverse sexual orientation, gender identity and expression (SOGIEs), increase challenges to fulfilling youth SRHR [20,21]. These factors limit the ability of youth to exercise bodily autonomy or access SRHR services in non-disaster times. Consequently youth experience poor SRHR outcomes including unmet need for modern contraception, and high rates of sexually transmissible infections, sexual- and gender-based violence, and early and unintended pregnancies [22].

Following climate-related disasters in Fiji and Tonga, increased rates of sexual- and gender-based violence including marital rape, sexual exploitation, and forced sex work were reported [23,24]. However, a clear understanding of the scale and age groups affected remain absent due to limitations in age- or sex-disaggregated data. Reports noted reluctance from

communities to disclose SRHR issues due to stigma, and a tendency to normalise and minimise sexual- and gender-based violence [23,24]. Further, the intersections of gender inequity, poverty and SGBV increased SRHR risks for women, girls, and sexual and gender minorities [8,23,24]. More recently, a scoping review of the humanitarian response to the 2020 Tropical Cyclone (TC) Harold in Fiji, Vanuatu, and Tonga identified limited focus on youth, and insufficient disaggregated data to understand youth SRHR needs [25]. Importantly, gender inequity, stigma, and unilateral decision-making by community power holders affected SRHR service provision during TC Harold responses [25]. Given the strong influence of sociocultural and structural determinants on SRHR across the disaster risk management cycle (preparedness, prevention, response, and recovery), understanding youth SRHR risks in disasters requires a holistic approach that examines these multi-level factors.

Globally, political, cultural, religious and social factors not only restrict individuals' abilities to realise SRHR, but also shape institutional capacity and willingness to provide SRHR services [26]. The influence of multi-level political, sociocultural, and religious factors on SRHR are an example of political intrusion into public health, resulting in available services not meeting the complex SRHR needs of youth [26,27]. Addressing youth SRHR requires an intersectional understanding of how social systems, power dynamics, and identity influence youth SRHR experiences, outcomes, and service provision. This includes identifying opportunities for integrated approaches that address youth SRHR risks before, during, and after disasters [19,28,29].

Local community-based service provision models that focus on holistic SRHR services increase accessibility for youth in non-disaster settings [27,30]. These models provide information and increase SRHR knowledge, address multi-level factors related to unequal power dynamics, sociocultural norms, and government policies that challenge youth SRHR, and facilitate access to clinical services. Community organisations are often the main sources of SRHR information and support for youth who are marginalised from mainstream services [27,30]. Despite their potential, these efforts by grassroots community organisations to transform norms, values, bias, and stigma that limit youth SRHR remain under-represented in peer-reviewed literature [26,27].

Moreover, the diversity and value of community-driven efforts are largely missing in research and practice within the humanitarian sector. While humanitarian needs are increasing globally, local community organisations receive limited funding and support and remain under-utilised in humanitarian responses [31]. The need for localisation of humanitarian responses has been enshrined in global commitments for sustainable development and disaster risk reduction efforts [31–33], but has remained a contentious issue in practice [34,35]. Issues of power and trust and divergent understandings of localisation impede collaborative partnerships, highlighting systemic and structural challenges to effective humanitarian action [34,35]. Recognising the need for just and radical action, communities and humanitarian organisations have called for transformative changes to current practices. This includes flexible responses, reducing organisational carbon footprints, and prioritising local community engagement at all levels [1]. Local action is especially important in the Pacific, as the remote and dispersed geography of the region mean local organisations and community members are often first responders in disasters [9]. To ensure Pacific youth SRHR risks are not increased in disaster contexts, it is critical to understand the role of community organisations in addressing youth SRHR.

## 1.2 A capitals approach to addressing risks to youth SRHR in disasters

The Recovery Capitals Framework (Box 1) draws on the Community Capitals concept to define capitals as "resources that can be maintained, increased and drawn upon to support wellbeing" [[36] page 56]. It considers how seven capitals–natural, built, political, cultural,

human, social, financial–intersect with each other, at individual, households and community levels, collectively influencing well-being after disasters [36]. We used the Recovery Capitals Framework to guide the analysis of our research findings so we could understand the multiple factors, from the individual to the structural levels, that influenced youth SRHR in a disaster context.

---

### Box 1. Definition of the seven capitals from the Recovery Capitals Framework [36]

1. **Natural capital**: Refers to the natural environment, geographical location, resources, and overall health of ecosystems

2. **Built capital**: Involves the design, building and maintenance of physical infrastructure, such as hospitals and evacuation centres

3. **Political capital**: Involves issues related to power, equity, agency, voice, and inclusion, as well as governance, leadership, and policy

4. **Cultural capital:** Reflects the way people understand and know the world, and how they act within it

5. **Human capital:** Refers to people's skills, knowledge, and capabilities, including ability to access resources and information

6. **Social capital**: Reflects connections, reciprocity, and trust among people and groups

---

Social capital is an important tool and strong influencer of health in disaster risk management, supporting community preparedness, response, and recovery from disasters [37]. Referring to connections, reciprocity, and trust at individual and community levels, there are three types of social capital; 1. bridging (strong ties between a close group of people), 2. bonding (looser ties between a broader group across intersections of race, gender, sexuality), and 3. linking (ties connecting people with powerholders and decision-makers) [38]. Social capital in the form of informal networks and trusted community organisations play a key role in navigating stigma and discrimination to increase access to SRHR services in disasters, particularly for women and those with diverse SOGIEs [39–41]. Stigma and shame related to cultural capital can restrict access to support, and cultural perceptions and social hierarchies influence people's decisions about which information sources they trust and which services they access [39,42]. In these contexts, social support networks are critical for mobilising resources and providing trusted information for women and people of diverse SOGIEs in disaster contexts [39–41]. A capitals perspective provides opportunities to identify these nuances within communities, and to understand how community organisations navigated challenges to identifying and addressing youth SRHR. To our knowledge, youth SRHR risks in the Pacific have not yet been examined through a multiple capitals framework like the Recovery Capitals Framework.

Community organisations and networks are experts in their contexts and are key to addressing the multi-level factors that affect youth SRHR [28]. There is a gap in the literature related to their experiences of navigating challenges and drawing on existing networks to provide services in disaster contexts. This research aimed to address this gap by exploring the

experiences of community organisations and networks following TC Harold in Fiji, Vanuatu and Tonga. The objectives were to understand how they identified and responded to youth SRHR needs in the disaster responses, and to highlight strengths in how they navigated current challenges to youth SRHR in disaster response and recovery.

## 2 Methods

### 2.1 Study setting and design

This study was conducted remotely from Melbourne, Australia, with participants in Vanuatu, Fiji and Tonga. In 2020, TC Harold made landfall in the Pacific around the same time as when COVID-19 was declared a pandemic, causing significant damage particularly in Fiji, Vanuatu and Tonga, with humanitarian responses triggered in all three countries [43]. According to the World Risk Index 2021, Vanuatu has the highest disaster risk worldwide, with Tonga having the third highest risk, and Fiji ranked at 14 [11]. All three countries have experienced category 5 cyclones (the strongest tropical cyclones) that appear to be increasing in frequency [10]. Prior to 2020, the three most recent category 5 cyclones to affect these countries were TC Pam (Vanuatu, 2015), TC Winston (Fiji, 2016) and TC Gita (Tonga, 2018). In 2020/2021 alone, the region experienced three severe tropical cyclones (TC Harold in April 2020, TC Yasa in December 2020, and TC Ana in January 2021), highlighting the increase in frequency and severity of climate-related disasters in the region.

We conducted a qualitative study using key stakeholder interviews, and our reporting is guided by the consolidated criteria for reporting qualitative research (COREQ) [44].

### 2.2 Reflexivity and theoretical underpinnings

This research was envisaged when the primary researcher (NM) was working in an international non-governmental organisation for SRHR and collaborating with Pacific Member Associations while based in Australia. NM has a professional background in clinical emergency medicine, community development work in SRHR, and personal experience of growing up in an island nation. Guided by Paulo Freire's theory of praxis [45] which recognises the importance of considering sociocultural and political implications of research and a commitment to challenging the status quo, we approached this research using theoretical standpoints that prioritise social justice and human rights [46,47]. This included recognising youth as a diverse group, with multiple intersecting identities that may influence their SRHR experiences, acknowledging that service provider experiences will be influenced by multiple individual, community, and structural factors, and examining ways to transform existing power dynamics to advance youth health and rights in disasters. To understand existing strengths within community organisations, we used the Recovery Capitals Framework as a strengths-based lens through which to view our findings [36]. The Framework allows us to recognise the dynamic, interlinked nature of disaster risks for youth SRHR, and identify inclusive, community-focused approaches with a focus on equity and social justice [36].

### 2.3 Participants, recruitment, and sampling

Participants were eligible for inclusion if they were involved in provision of SRHR or humanitarian information, support, or services to communities in Fiji, Vanuatu and Tonga following TC Harold, able to participate in an interview in English, had internet access and ability to engage in digital and remote interviews given COVID-19 restrictions at the time of data collection. Participants were recruited using several methods including distribution of a flyer via social media, e-mailing Pacific and international development and humanitarian organisations

directly, using professional networks of the research team, and snowball sampling, where participants recommend contacts through their networks [48]. Those who fulfilled the inclusion criteria were invited to participate in a semi-structured individual interview.

## 2.4 Data collection and management

Data were collected between September 2021 and January 2022 by NM. Due to ongoing risks and restrictions related to the COVID-19 pandemic, the research team was in Melbourne, Australia, and the study was conducted remotely using online individual qualitative interviews. All participants who expressed interest were interviewed using semi-structured individual qualitative interviews over Zoom or telephone, with one interview conducted asynchronously over e-mail [49,50]. Interviews were conducted in English and lasted between 45 to 75 minutes. An interview guide was piloted with a humanitarian SRHR service provider based in Fiji, and no changes made after the pilot (S1 Appendix). The guide consisted of 5 broad sections exploring the experiences of community organisations, networks, and humanitarian response organisations during TC Harold responses: 1) organisation and types of services provided, 2) any identified SRHR needs, 3) experiences of addressing youth SRHR, 4) experiences of engaging with youth, and 5) experienced challenges and potential opportunities to strengthen a focus on youth SRHR in future. Participants received mobile data vouchers (equivalent to AU $20) to reimburse their internet data use during interviews. We interviewed all 16 service providers who expressed interest, because we wanted to respect participant autonomy and value their time and interest. All interviews conducted over Zoom or telephone were audio-recorded and transcribed verbatim by NM, aided by an automated transcription service (Otter.ai) [51]. The e-mail interview followed the same question and answer format which we directly transferred to a Word document and did not require further transcription.

## 2.5 Data analysis

Guided by our theoretical standpoints and aims to identify opportunities for transformative change, data analysis was a combination of reflexive thematic analysis and framework analysis [52]. First, we analysed all transcripts inductively, staying close to participant experiences, identifying codes of meaning, and grouping codes around central concepts, or themes. To ensure the identified themes reflected participant experiences, a Padlet board of themes [53] was shared with all participants. Padlet is a free digital board that facilitates online collaboration, and all participants were provided with the link to access the board, with options to comment, give feedback, or upvote/downvote individual themes [53]. We then mapped the generated themes to the Recovery Capitals Framework [36], identifying challenges and strengths across the seven capitals. This process enabled us to explore how participants drew on existing strengths to identify and address challenges for youth SRHR.

## 2.6 Ethics and consent

Ethics was obtained from University of Melbourne Human Ethics Committee (Ref 2021-20443-20901-3) and all research was conducted in accordance with the National Statement on Ethical Conduct in Human Research guidelines [54]. Ethics approval from University of Melbourne was deemed appropriate by the research team and advisors in the Pacific, as all interviews were conducted remotely from Melbourne with participants from three countries, focusing on their experiences and perspectives from reflecting on a disaster response. Participants were also asked whether permission was required from their respective organisations before data collection. Written consent was obtained over e-mail prior to interview participation, participants were provided with the opportunity to ask questions over e-mail or Zoom,

and verbal agreement to participate was reiterated prior to commencing interviews. All participants were given pseudonyms using names commonly used in the three respective countries to maintain confidentiality.

## 3 Results

We conducted 15 interviews with a total of 16 participants (one was a joint interview with two participants), who were diverse in their ages, gender, and locations (Table 1). All participants were from community-based non-governmental organisations, civil society organisations, and community networks. Many had multiple paid or volunteer roles and experiences working across several of the organisations.

Participants described interconnected factors across all seven capitals (Box 1) that challenged youth SRHR service provision and access. In the following sections, we discuss how participants drew on social capital to navigate challenges from all other capitals. We focus on social capital as this was central to how participants identified and addressed challenges to youth SRHR. Social capital as a key strength was evident during data collection and analysis by NM and confirmed through discussions with the co-authors from the Pacific (TR and AD). We highlight potential opportunities to harness and build on existing strengths in social capital to address youth SRHR in future.

Participant accounts demonstrate how interconnected factors across several individual, community, and structural levels affect ability to identify and address youth SRHR in disasters. Fig 1 summarises the interactions between capitals and the different ways participants drew on social capital to address challenges related to the other six capitals of the Recovery Capitals Framework [36].

**Table 1. Sociodemographic characteristics of participants.**

| | Participants (n = 16) |
|---|---|
| Participant location | |
| Fiji | 7 |
| Vanuatu | 7 |
| Tonga | 1 |
| New Zealand (representing Vanuatu) | 1 |
| Age | |
| 20–29 | 3 |
| 30–39 | 5 |
| 40–49 | 2 |
| 50–59 | 5 |
| 60+ | 1 |
| Types of organisations represented by participants† | |
| SRHR organisations including clinical service provision | 4 |
| National societies of international disaster response organisations | 3 |
| Pacific regional organisation | 1 |
| Youth-led or youth-focused networks | 3 |
| Women's rights organisations | 4 |
| LGBTQIA+ organisations | 2 |

† Some participants represented more than one organisation.

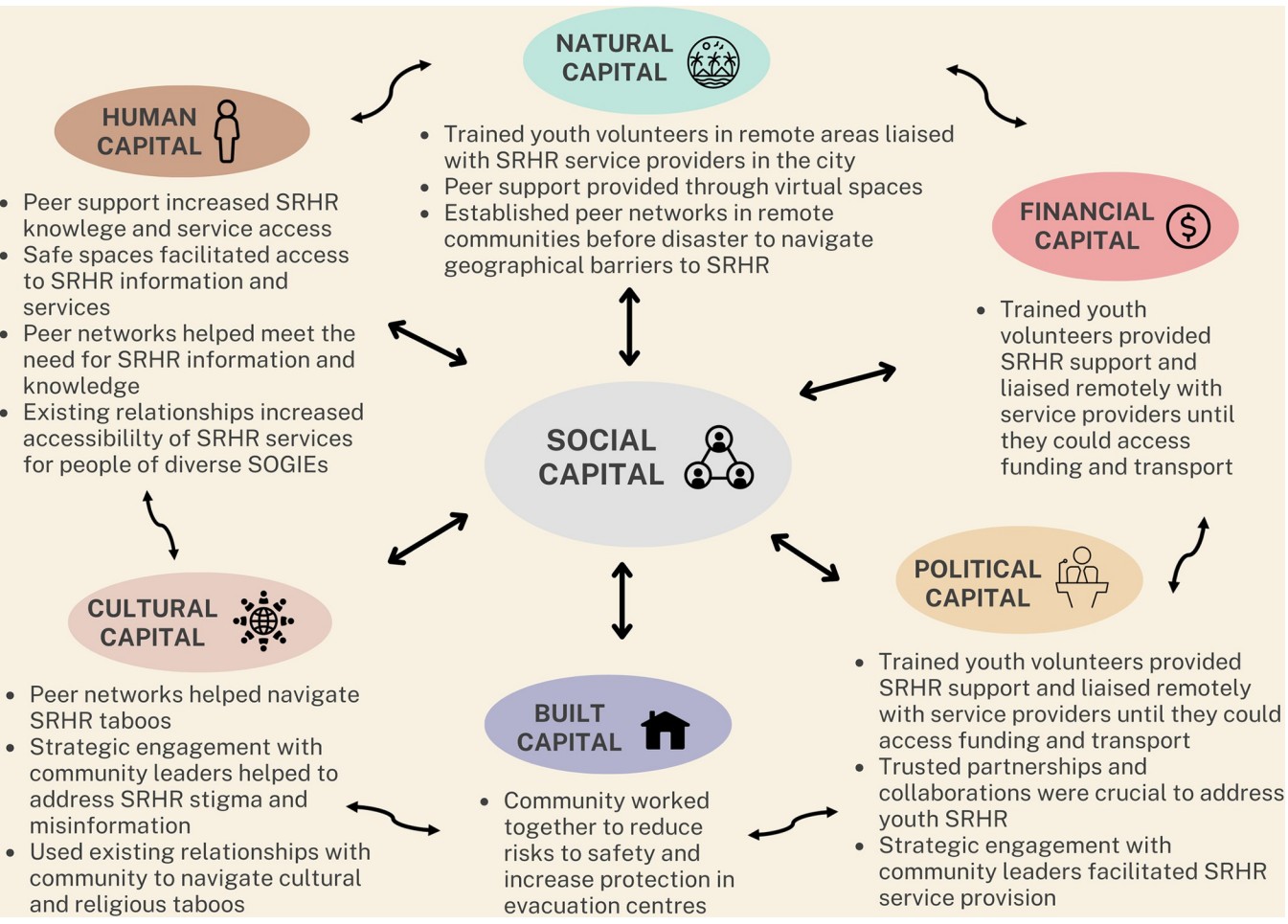

**Fig 1. How community organisations and networks utilised strengths in social capital to mitigate challenges to youth SRHR.** The social resources participants drew on are summarised through how they helped address challenges related to human, built, financial, political, natural, and cultural capitals. *Adapted from the Recovery Capitals Framework [36].*

We discuss our findings below, using illustrative quotes to demonstrate existing strengths in social capital. We present five ways in which social capital helped address youth SRHR; (1) youth networks and trained volunteers, (2) responding to identified needs, (3) inclusive and strategic community action, (4) existing relationships with communities, and (5) trusted partnerships and collaborations.

## 3.1 Social capital as a key strength to address challenges

Community organisations and networks demonstrated multiple ways in which they harnessed and built on different types of bridging, bonding, and linking social capital to identify and address youth SRHR in the context of disasters. Social capital was a critical resource to mitigate challenges related to all other capitals.

**3.1.1 Youth networks and trained youth volunteers.** Organisations with a network of trained youth volunteers described how the network facilitated identification of youth SRHR needs following TC Harold. These networks were essential to address challenges related to natural, political, and financial capitals. For instance, in Vanuatu trained volunteers in rural areas liaised remotely with services in the capital, reporting on SRHR needs and providing support

until clinical service providers could reach the islands. One participant, Matiu, explained that challenges in accessing funding and transport to remote areas made it impossible to reach affected populations within the required 72-hour period. He described how youth networks mitigated the possible risks of these delays, as below:

*"Cyclone or any disasters can hit anywhere where we don't have presence, so we work through those [youth] networks . . .we need to be [at] the scene within 72 hours but in most cases that's not possible, logistically. So we work with those [youth networks] to help us. . . with information and . . . data [on] what's happening, and then we do the preparations in Vila [city] before we get there [to remote communities]"*–Matiu, Vanuatu

Similarly, Nia, a participant from Fiji, described how being stranded in a remote island due to multiple disasters enabled their youth peer educators to establish relationships, and address youth SRHR needs that may not have been evident if they were only there for the humanitarian response period. Over the two months they were there, youth peer educators facilitated access to information and services including SRHR counselling, STI testing and treatment, and met contraceptive needs. Nia's account below highlights how important social capital is to facilitate youth SRHR access, and how it builds over time.

*"Our last [humanitarian] response. . .we were stranded in the island for two months. Because [while] we were there two other hurricanes came in. And the youth that we brought in they started playing volleyball with youth in the community . . .and they [came to the clinic] with somebody [who] was complaining of having painful urination. . . and then [we] do the counselling. . . . [through youth volunteers] mingling with their peers . . . [youth were] coming up one by one [for SRHR services]. . . It's very important to engage the youth so that they can identify their own group, and also identify the problems that is within."*–Nia, Fiji

Participant accounts demonstrated how social capital in the form of youth volunteers and networks helped address challenges related to geographical remoteness, difficulty accessing funds on time, and time restrictions related to humanitarian response funds. The challenges described indicate natural, political, and financial capitals.

**3.1.2 Responding to identified needs.** Participants drew on social capital to meet community needs that were identified during responses, thereby mitigating challenges to SRHR information and service access. For example, where SRHR organisations and services were not accessible to remote regions, youth networks adapted to provide crucial social support through innovative methods. One participant, Levani, described efforts to create virtual 'safe spaces' for youth who have experienced trauma following disasters. These spaces enabled access to peer support, and they used confidential 'phone tree' systems to facilitate referrals to clinical and counselling support. Levani explained how important these networks were especially for those in remote and rural areas, who do not regularly have access to SRHR information, counselling, or support services.

*"We have been able to convene some . . . safe. . . spaces [for] young people who've been traumatised or have been impacted by the disasters. . .building in a network of peers that can create social support systems at the community level . . .and this is in particular for areas that are in rural remote and maritime areas . . . because of the remoteness, but also because of . . . the lack of communication like internet so we know they are unable to connect with someone on mainland."*–Levani, Fiji

Previous experiences of responding to disasters within their communities helped participants adapt their service provision based on identified needs. As one participant, Alani, discussed below, her organisation identified a need for more SRHR information and support in rural areas during a previous response, which motivated them to form a peer network within the remote province. She discussed how consulting with communities ensures they continue to provide services that meet community needs. These networks were crucial for provision of peer support, counselling, and referrals to SRHR services during TC Harold responses.

*"So we [found] out [that] a lot of this [SRHR] information, we are taking it for granted in Port Vila [the city] [but] it's needed where we went and responded. So, then we thought okay, I will put in our network there. . .We've seen the high need. . .so we've [been] making sure that anything [we provide] we've asked them what do you want? . . . They were really vocal about what they want. But no one [had] ever asked them"*–Alani, Vanuatu

Accounts demonstrated how participants used social capital, through provision of online support networks and establishment of peer networks within remote areas, to address identified community needs. Drawing on social capital helped them navigate the challenges related to natural and human capitals, such as geographical barriers to SRHR information and services. Utilising social capital through community engagement and consultation also helped ensure services are meeting the needs of communities.

**3.1.3 Inclusive and strategic community action.**   All participants identified SRHR and youth SRHR issues as taboo subjects in many of their contexts, highlighting how existing challenges to realisation of SRHR are exacerbated in disasters. Taboos associated with talking about contraception, menstrual health, and other SRHR issues intersected with taboos related to age. Consequently, participants explained that even if youth had SRHR needs, the compounding taboos stop them from being able to access services. Josefa discussed navigating these challenges by being strategic about how they approached and involved village headmen and community leaders when addressing SRHR issues after disasters.

*"You can't just walk into a village hall and start distributing condoms because in the next five minutes they will start chasing you out of the village [laugh] . . . we need to go about strategic ways of how we can talk to our village headmen and get [community leaders] to be involved when disseminating this information. [We] provide contraceptives and give them information. . . as to why this is good, why this is bad, why preventative mechanism should be used. These are all regarded as taboo especially for Fijians"*–Josefa, Fiji

Many participants in Fiji expressed hope that mindsets are changing and noted how communities were working together to address safety and protection issues, particularly in evacuation centres. Participant descriptions exemplified how community social capital helped mitigate risks of sexual- and gender-based violence. For instance, community members accompanied and protected each other to facilitate safe access to water, sanitation, and hygiene (WASH) facilities, especially if they were located outside. Participants described community efforts to ensure no one had to travel to WASH facilities alone or in the dark, to minimise risks of violence for women, people with disabilities, and those with diverse SOGIEs.

Alani from Vanuatu explained how community organisations in Vanuatu created safe spaces where youth and adults could discuss SRHR issues in confidence and support each other during disaster responses. Creating networks between different provinces for sharing experiences and information supported youth to navigate taboos related to SRHR.

*"[SRHR issues are] very sensitive issues. And so we have to deal with it behind closed doors. So that safe space [that] we are creating, that's where those dialogues are happening. . .in the village [SRHR issues are] taboo, [we] don't talk about it, you get punished. . . [During TC Harold responses] we've taken [trained youth peers] up the north province [to] help us [with the response]. . . it's locally-led which means they share their own experience. . .the community [is] like oh, we can connect with that"*–Alani, Vanuatu

Participant accounts highlighted how social capital was used to bridge challenges to political capital, by strategically engaging with community leaders to facilitate SRHR information and service provision. Participants used social capital in the form of virtual and physical peer networks to connect youth between different provinces, enabling them to support each other to access SRHR information and services within safe spaces. They also discussed how community action helped reduce SRHR risks in evacuation centres, illustrating how social capital mitigated risks related to built capital.

**3.1.4 Existing relationships with communities.** Established relationships with community members before disasters helped participants facilitate delivery of youth SRHR services during COVID responses. As Ana explained, her organisation was already engaging with youth in Tonga, delivering youth peer education programs, and providing information and training on SRHR issues such as sexual- and gender-based violence. This made it much easier for youth and community members to access their service during disasters as the community already knew the organisation.

*"In stable times, [our organisation] already have established stakeholders in these islands. And we already run workshops on peer education for these young people, we also train them on linking COVID-19 to sexual [and] reproductive health, we train them on the education and life skills, gender-based violence training. We have [our] youth drama group. So we have visited them before. And when the crisis hit, we are no strangers for the community and. . . it's a bonus for the accessibility of the services."*- Ana, Tonga

Established relationships were also crucial to address the taboos mentioned previously, especially with respect to cultural or religious stigma. Culturally respected leaders drew upon their existing relationships within their communities to overcome some of the stigma associated with youth SRHR. For instance, community organisations who have affiliations with their local church addressed taboos through having discussions with church communities. This important use of social capital to address and enhance cultural capital was exemplified by participants when explaining their strategies to mitigate the risks to youth SRHR in patriarchal traditional communities. As one participant, Helen, explained:

*". . .Because of their church affiliation, they [community organisations] know people everywhere. So wherever they go, immediately they've broken down the barrier. And [compared to international NGOs] if small groups can be doing this work, I think the impact [on community attitudes and knowledge] is greater. . .it really is amazing to observe when you see local to local, so whether it's Vanuatu or Fiji or wherever, I think that's where the magic happens. . . I think that's where the power is."*–Helen, Vanuatu

Existing relationships were especially important to address the diversity of youth SRHR needs. A participant in Fiji, Litia, explained that as a trans woman she is known and trusted within sexual and gender diverse communities, which made it easier for her organisations and networks to deliver services for sexual and gender diverse youth during TC Harold responses.

*"Whenever we were deployed to the field . . .I was representing the transgender community, and as well the LGBT community"*–Litia, Fiji

Participant descriptions illustrated how social capital from existing relationships were drawn upon to address challenges to cultural capital such as social and religious taboos, thereby increasing access to SRHR services for youth with intersecting SRHR needs.

**3.1.5 Trusted partnerships and collaborations.**   Trusted partnerships between community organisations and networks allowed participants to integrate SRHR responses within their work. This included establishing thematic working groups across gender, health, human rights, and climate justice, and working in collaboration on policy advocacy and service provision before, during, and after disasters. One participant, Talei, described building youth collectives where they trust each other and share information. Social capital through youth collectives enables community organisations and networks to identify opportunities for continued collaborations to advance youth health and rights.

*"This is about a youth collective, it's about youth movement building. . . we become friends with these people who are also our colleagues. . . we have now established that trust [and] open communication, and we've worked on several other things out of these partnerships as well."*–Talei, Fiji

Participants involved in the TC Harold formal humanitarian responses recognised positive outcomes from collaboration and training between the WASH cluster and the Gender cluster. They described how it facilitated cohesive distribution of menstrual products and increased awareness and knowledge on menstrual health management.

In contrast to the collaborations within formal disaster response clusters, participants from community organisations expressed frustration with current practices. They stated that local NGOs often feel disrespected as they are marginalised or excluded from formal humanitarian responses, despite having local knowledge and expertise. Participants emphasised the importance of collaborating with local organisations, who have been working with their communities for many years, particularly in addressing SRHR related challenges. One participant, Fiva, explained the frustration many local organisations feel when their ongoing work with communities are not valued or acknowledged within humanitarian responses. She advocated for respectful collaboration and partnerships, as described below:

*"When international humanitarian actors are coming in, they're encouraging and supporting the [international] INGOs, but not the local NGOs. And the local NGOs, we're not feeling threatened, we're just feeling that it's lack of respect. When you're flying in from somewhere and you're going past us and you're going directly to the community that we've been working with for years and years. . . that's not respect, and I really hope that when disasters happen that we could have more local NGOs, local actors, engaged to participate in . . . humanitarian responses."*- Fiva, Vanuatu

Fiva also emphasised the importance of recognising the work that organisations are doing with youth and the potential for sustainable change through localised efforts. She described how her organisation builds the capacity of youth to engage with community leaders, raising awareness about SRHR issues and focusing on addressing root causes of sexual and gender-based violence.

*"But also look at what we've done [working with youth]. . . we've trained [unemployed youth] and we sent them to the communities for six months and [they] made a huge impact. They*

*influenced the chiefs and key community leaders. This is what we want to see. We want to localise those efforts. And once communities are empowered, you can imagine the rest of the things that they will do. To bring . . . social, climate and economic changes to our communities."*–Fiva, Vanuatu

Partnerships between community organisations and networks were important avenues through which participants built youth movements based on trust and reciprocity, thereby increasing social capital to address youth SRHR needs. Participants used social capital through trusted partnerships to mitigate challenges related to political capital such as SRHR not being prioritised within formal humanitarian responses. Participants advocated drawing on the existing strengths within local organisations to improve collaborations between international and local organisations and ensure youth SRHR needs are identified and addressed.

## 4 Discussion

We explored the experiences of community organisations and networks in identifying and addressing youth SRHR following TC Harold in Fiji, Vanuatu and Tonga. We illustrated how participants drew on existing strengths—particularly social capital—to address challenges across the multiple factors affecting youth SRHR (Fig 1). Using the Recovery Capitals framework, our findings show how existing challenges such as limited access to services, and cultural or religious taboos, were exacerbated in disaster contexts and affected youth SRHR in multiple ways. Social capital was a crucial resource through which participants navigated challenges related to natural, human, political, built, financial, and cultural capitals.

Participant accounts emphasised that those in rural and remote areas experienced compounding challenges to their SRHR, due to the geographical limitations in natural capital. Limited access to SRHR information and services pre-disaster in remote areas, an example of limited human capital, were exacerbated by delays in funding and transport to deliver timely SRHR services during humanitarian responses, which signify challenges in political and financial capital. Faced with these decreased natural, political, human, and financial capitals, participants drew on and further developed their social capital to address these challenges. This included convening virtual safe spaces to increase access to SRHR information and services for youth, providing bridging, bonding, and linking social capitals. The work of community organisations pre-disaster, such as training youth volunteers within remote areas, enabled timely responses to identify youth SRHR issues and provide immediate support. Youth volunteers provided essential linking social capital by collaborating with services in the city to ensure services met identified needs when they were able to access the remote areas. Participant descriptions also emphasised the importance of social capital through trusted peer interactions to address SRHR needs that develop over time when displacement and recovery processes are prolonged. This echoes previous research highlighting that social capital is critical to minimise the immediate impacts of SRHR service disruptions [39].

Participants described deeply ingrained cultural and religious taboos and traditions that continue to limit the ability of youth to seek SRHR information and services. This supports recent literature highlighting how existing norms and practices increase SRHR risks in disasters, and the importance of addressing these factors before disasters occur [8,55]. As participants have a nuanced understanding of the contexts they live and work in, they could leverage existing relationships to navigate challenges related to cultural capital respectfully and effectively. Organisations and networks who have existing trusted relationships with their communities demonstrated an intersectional understanding of the diversity of youth SRHR needs. They were especially important to increase access to SRHR information and services for youth

from marginalised populations, including youth with sexual and gender diversity, single mothers, and sex workers. Social capital and community-based services are increasingly being recognised as crucial in ensuring access to SRHR information and services, especially for those who are marginalised [27,39].

## 4.1 Implications for policy and practice

Our findings have important implications for policy and practice, which we summarise as three main points. First, existing relationships, trusted collaborations, and peer networks are crucial for identifying and addressing youth SRHR. As highlighted from Fig 1, influences on youth SRHR are complex and intersecting, emphasising the need for integrated and holistic approaches to address SRHR in the context of disasters [26,28]. Participants also illustrated how youth SRHR needs evolve over time following a disaster, and the importance of having trusted networks and services beyond the specified timeframes of a humanitarian response to meet their needs. In humanitarian contexts, marginalised or minoritized groups face greater barriers to access to basic services, especially with respect to SRHR [56,57]. Consequently, the strong influences of socio-cultural and political factors on access to SRHR information and services for Pacific youth increase their risks of worse outcomes in disasters. As a diverse population group with changing and intersecting SRHR needs and experiences, safeguarding youth SRHR is critical to ensuring the well-being of Pacific youth in disaster contexts. Strengthening and formalising the peer networks and safe spaces provided by community organisations have the potential to enable youth SRHR needs to be identified and addressed within the humanitarian response architecture. This would enable timely and cohesive responses to reduce youth SRHR disaster risks.

Second, our findings demonstrate how leveraging strengths in community social capital can support localisation of humanitarian responses. As evident from Nia's account of being stranded by multiple cyclones (section 3.1.1), the Pacific already face multiple hazards and complex disasters, with communities not having time to recover from one crisis before the next one occurs. In the contexts of increasing humanitarian needs and limited funding globally, community organisations and networks are key for identifying and responding to youth SRHR needs. Already working at the nexus of humanitarian responses and development work, community organisations are utilising their social capital to provide innovative solutions that meet the needs of the communities they work with. Recognising local expertise and increasing flexible funding and resources for these partnerships would ensure they are able to scale up their work. This is supported by recent research from Fiji, Vanuatu, and the Solomon Islands, advocating for flexible funding that allows community organisations to adapt to evolving needs during disasters [9]. Our findings highlight how the work of community organisations to increase SRHR knowledge and capacity within their networks and communities before disasters provides contextually relevant services that can be sustained before, during, and after disasters.

Finally, sustainable approaches to disaster risk reduction in this multi-hazard region require the complex intersections of capitals to be understood and used strategically. To achieve this, genuine partnerships between international humanitarian organisations, governments and local organisations are crucial. The strengths demonstrated and described by participants present opportunities for collaborations that bridge international and local expertise, investing in existing social capital to enhance and transform other capitals. This process, known as 'spiraling up', has been demonstrated in other settings, where investing in human capital resulted in a flow of resources or capacities across political, financial, cultural, and social capitals [58]. To illustrate this spiraling-up potential in our findings, we refer to the example by Fiva (section

3.1.5) who described her organisation's work with unemployed youth. They trained unemployed youth, thereby increasing human capital, and supported their engagement with community members over a period of time, increasing community social capital. She reported youth influencing and changing mindsets of community leaders, highlighting how drawing on human and social capital can increase cultural and political capital. She goes on to emphasise how these localised efforts can result in transformative changes for their communities.

## 4.2 Limitations and strengths

This is the first study in the Pacific to apply the Recovery Capitals framework to understand how community organisations address disaster risks for youth SRHR. The main challenge for this study was that remote interviews limited the number of participants, as stakeholders were busy responding to COVID-19 and multiple climate-related disasters during the data collection period. However, a key strength was that remote interviews enabled us to interview participants from three countries. Despite only being able to interview one participant from Tonga, the process provided valuable insights into common factors affecting youth SRHR in disasters from three different countries in the Pacific. Our study contributes a strengths-based understanding of the factors that influence youth SRHR in disaster contexts. Drawing on the lived experiences of community organisations and networks we have highlighted important implications for policy and practice.

## 5 Conclusions

As a large youth population facing increasing climate-related disasters, protecting and promoting the SRHR of Pacific youth is essential for securing healthier and more resilient societies in the region. Our study emphasises the crucial role of social capital in identifying and addressing youth SRHR in disaster contexts. We demonstrate how community organisations and networks are using existing strengths in trusted partnerships and collaborations to navigate challenges to youth SRHR strategically and effectively. Pre-existing challenges to youth SRHR, such as geographical barriers to SRHR information and access, cultural and religious taboos, and limited SRHR knowledge, were compounded by disaster-specific challenges such as limited funding for timely and effective responses to meet youth SRHR needs. Exploring the multiple influences on youth SRHR service provision and access in the Pacific through a capitals framework provided crucial insights to inform strategies for reducing youth SRHR risks before, during, and after disasters. Our findings highlight the essential role of community organisations and networks in addressing youth SRHR in disasters. Understanding how participants used social capital to navigate challenges within natural, political, cultural, human, built, and financial capitals highlights opportunities for 'spiraling-up' of capitals, through harnessing the existing strengths of community organisations. Supporting the localisation of humanitarian responses in the Pacific is crucial for transformative advancement of the SRHR of Pacific youth.

## Supporting information

**S1 Appendix. Interview guide.**
(DOCX)

## Acknowledgments

We are thankful to the participants who were so generous with their time, and in sharing their experiences. We also thank colleagues in Australia and the Pacific who helped distribute recruitment information and facilitate connections.

## Author Contributions

**Conceptualization:** Nabreesa Murphy, Peter Azzopardi, Meghan A. Bohren.

**Data curation:** Nabreesa Murphy.

**Formal analysis:** Nabreesa Murphy, Phoebe Quinn.

**Methodology:** Nabreesa Murphy, Peter Azzopardi, Meghan A. Bohren.

**Project administration:** Nabreesa Murphy.

**Supervision:** Peter Azzopardi, Kathryn Bowen, Meghan A. Bohren.

**Writing – original draft:** Nabreesa Murphy.

**Writing – review & editing:** Nabreesa Murphy, Peter Azzopardi, Kathryn Bowen, Phoebe Quinn, Tamani Rarama, Akanisi Dawainavesi, Meghan A. Bohren.

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
