## [Decision Letter · Decision Letter 0]

18 Jan 2023

PGPH-D-22-02067

Using social capital to address youth sexual and reproductive health and rights in disaster preparedness and response: a qualitative study highlighting the strengths of Pacific community organisations and networks

Dear Dr. Murphy,

Thank you for submitting your manuscript to PLOS Global Public Health. After careful consideration, we feel that it has merit but does not fully meet PLOS Global Public Health’s publication criteria as it currently stands. Therefore, we invite you to submit a revised version of the manuscript that addresses the points raised during the review process.

There are a few minor issues that the reviewers have pointed out which needs to be addressed and I invite you to revise your manuscript based on the comments by the reviewers. 

We look forward to receiving your revised manuscript.

Kind regards,

Mathew Sunil George

Academic Editor

Journal Requirements:

3. In the Funding Information you indicated that no funding was received. Please revise the Funding Information field to reflect funding received.

4. We have noticed that you have uploaded Supporting Information files, but you have not included a list of legends. Please add a full list of legends for your Supporting Information files after the references list. 

Additional Editor Comments (if provided):

Reviewers' comments:

Reviewer's Responses to Questions

**Comments to the Author**

1. Does this manuscript meet PLOS Global Public Health’s publication criteria? Is the manuscript technically sound, and do the data support the conclusions? The manuscript must describe methodologically and ethically rigorous research with conclusions that are appropriately drawn based on the data presented.

Reviewer #1: Yes

Reviewer #2: Yes

2. Has the statistical analysis been performed appropriately and rigorously?

Reviewer #1: N/A

Reviewer #2: N/A

3. Have the authors made all data underlying the findings in their manuscript fully available (please refer to the Data Availability Statement at the start of the manuscript PDF file)?

Reviewer #1: No

Reviewer #2: Yes

4. Is the manuscript presented in an intelligible fashion and written in standard English?

Reviewer #1: Yes

Reviewer #2: Yes

5. Review Comments to the Author

Reviewer #1: Authors should make excerpts of the transcripts relevant to the study available in an appropriate data repository, within the paper, or upon request if they cannot be shared publicly. If even sharing excerpts would violate the agreement to which the participants consented, authors should explain this restriction and what data they are able to share in their Data Availability Statement.

There should be some form of statement about the excertps of the transcripts. Options are listed below:

The datasets generated during and/or analysed during the current study are available in the [NAME] repository, [PERSISTENT WEB LINK TO DATASETS].

The datasets generated during and/or analysed during the current study are available from the corresponding author on reasonable request.

All data generated or analysed during this study are included in this published article (and its supplementary information files).

The datasets generated during and/or analysed during the current study are not publicly available due to [REASON(S) WHY DATA ARE NOT PUBLIC] but are available from the corresponding author on reasonable request.

Reviewer #2: This manuscript fills a gap in research associated with sexual and reproductive health needs and rights of adolescents in areas prone to natural disasters. Statistical analyses were not needed, as this qualitative data collection used appropriate qualitative data analyses to present their results. This research will serve as the basis for additional quantitative and quantitative research documenting SRHR among adolescents in fragile conditions, which should serve as the basis for building programs to mitigate these adolescents' challenges. This research clearly shows that partnering with local NGOs already working with adolescents will improve and strengthen approaches by international disaster relief organizations when they come in during disaster relief operations. This is a well-written manuscript which needs minor editing for a couple of typos (i.e.: P11, line 257; says "cased" which should be "cases". This research could have been strengthened by additional interview with people from Tonga--but the authors rightfully point out that this is a limitation.

6. PLOS authors have the option to publish the peer review history of their article (what does this mean?). If published, this will include your full peer review and any attached files.

**Do you want your identity to be public for this peer review?** For information about this choice, including consent withdrawal, please see our Privacy Policy.

Reviewer #1: No

Reviewer #2: **Yes: **Danuta Kasprzyk, MA, PhD

---

## [Decision Letter · Decision Letter 1]

3 Apr 2023

PGPH-D-22-02067R1

Using social capital to address youth sexual and reproductive health and rights in disaster preparedness and response: a qualitative study highlighting the strengths of Pacific community organisations and networks

Dear Dr. Murphy,

Thank you for submitting your manuscript to PLOS Global Public Health. After careful consideration, we feel that it has merit but does not fully meet PLOS Global Public Health’s publication criteria as it currently stands. Therefore, we invite you to submit a revised version of the manuscript that addresses the points raised during the review process.

We look forward to receiving your revised manuscript.

Kind regards,

Thu-Anh Nguyen

Academic Editor

Journal Requirements:

Additional Editor Comments (if provided):

Reviewers' comments:

Reviewer's Responses to Questions

**Comments to the Author**

1. If the authors have adequately addressed your comments raised in a previous round of review and you feel that this manuscript is now acceptable for publication, you may indicate that here to bypass the “Comments to the Author” section, enter your conflict of interest statement in the “Confidential to Editor” section, and submit your "Accept" recommendation.

Reviewer #1: All comments have been addressed

2. Does this manuscript meet PLOS Global Public Health’s publication criteria? Is the manuscript technically sound, and do the data support the conclusions? The manuscript must describe methodologically and ethically rigorous research with conclusions that are appropriately drawn based on the data presented.

Reviewer #1: Yes

3. Has the statistical analysis been performed appropriately and rigorously?

Reviewer #1: N/A

4. Have the authors made all data underlying the findings in their manuscript fully available (please refer to the Data Availability Statement at the start of the manuscript PDF file)?

Reviewer #1: Yes

5. Is the manuscript presented in an intelligible fashion and written in standard English?

Reviewer #1: Yes

6. Review Comments to the Author

Reviewer #1: Using social capital to address youth sexual and reproductive health and rights in disaster preparedness and response: a qualitative study highlighting the strengths of Pacific community organisations and networks.

Thank you for your well written, analyzed, and discussed research manuscript. I have some minor points that need to be revised.

Line 4-7 – please create two sentences out of the one sentence to break up your ideas.

Line 34- “increase risks” should be more specific. What are the increased risks?

Line 66 – Can you be more specific regarding the multi-level factors that challenge youth SRHR. It’s a broad statement and the sentences that follow don’t link to the multi-level factors.

Line 182 – you say all interviews were audio recorded and transcribed. However, earlier you say one interview was via email communication. Can you change the wording of this sentence as it was not all interviews.

Line 277 and several others – you say several times that participants described how social capital …… I am assuming that participants did not use the words social capital but in your analysis due to your framework you analyzed it to be social capital. So a change of wording in necessary in these sections were you say participants described. Something like participants described what we determined to be social capital.

Line 322-323 – This entire section how does it relate to climate change disasters. It seems very general and not related. Can you make some connections in this section.

Line 339 – “community members supported and enabled safe access to water, sanitation, and hygiene (WASH) facilities.” How? Be more specific with your examples. It will help to give the reader more context.

Line 445 - Trained them in what? Do you have more details/context.

7. PLOS authors have the option to publish the peer review history of their article (what does this mean?). If published, this will include your full peer review and any attached files.

**Do you want your identity to be public for this peer review?** For information about this choice, including consent withdrawal, please see our Privacy Policy.

Reviewer #1: No

---

## [Editor Report · Decision Letter 2]

14 Apr 2023

Using social capital to address youth sexual and reproductive health and rights in disaster preparedness and response: a qualitative study highlighting the strengths of Pacific community organisations and networks

PGPH-D-22-02067R2

Dear Dr Murphy,

We are pleased to inform you that your manuscript 'Using social capital to address youth sexual and reproductive health and rights in disaster preparedness and response: a qualitative study highlighting the strengths of Pacific community organisations and networks' has been provisionally accepted for publication in PLOS Global Public Health.

Best regards,

Thu-Anh Nguyen

Academic Editor
